# Vitamin D—The Iceberg in Endometriosis—Review and Meta-Analysis

**DOI:** 10.3390/jpm14010119

**Published:** 2024-01-20

**Authors:** Alexandra Ursache, Ludmila Lozneanu, Iuliana Elena Bujor, Cristina Elena Mandici, Lucian Vasile Boiculese, Alexandra Irma Gabriela Bausic, Mihaela Grigore, Demetra Socolov, Daniela Roxana Matasariu

**Affiliations:** 1Department of Obstetrics and Gynecology, University of Medicine and Pharmacy ‘Gr. T. Popa’, 700115 Iasi, Romania; alexandra.ursache@umfiasi.ro (A.U.); mihaela.grigore@umfiasi.ro (M.G.); demetra.socolov@umfiasi.ro (D.S.); daniela.matasariu@umfiasi.ro (D.R.M.); 2Department of Obstetrics and Gynecology, Cuza Vodă Hospital, 700038 Iasi, Romania; 3Department of Morpho-Functional Sciences I—Histology, University of Medicine and Pharmacy ‘Gr. T. Popa’, 700115 Iasi, Romania; ludmila.lozneanu@umfiasi.ro; 4Biostatistics, Department of Preventive Medicine and Interdisciplinarity, University of Medicine and Pharmacy ‘Gr. T. Popa’, 700115 Iasi, Romania; vasile.boiculese@umfiasi.ro; 5Department of Obstetrics and Gynecology, University of Medicine and Pharmacy ‘Carol Davila’, 020021 Bucharest, Romania; alexandra.bausic@drd.umfcd.ro

**Keywords:** endometriosis, vitamin D, 25(OH)D, 25(OH)D_3_, 1,25(OH)D, 1,25(OH)_2_D_3_, vitamin D binding protein, vitamin D receptor, biomarker, ELISA

## Abstract

(1) Background: Although vitamin D has many known biological effects, very little research has been conducted on how vitamin D may be related or play a role in endometriosis. The aim of our study was to perform an evaluation regarding vitamin D levels and possible implications in endometriosis through a statistical analysis of the data collected from the included studies. (2) Methods: For this review, we searched the Cochrane Central Register of Controlled Trials (CENTRAL), Web of Science, and PubMed/Internet portal of the National Library of Medicine databases using several keywords related to our topic. (3) Results: Only nine articles were identified as complete or possessing the capacity to compute all available data. We totalized a number of 976 patients with endometriosis and 674 controls. From the nine studies included in our analysis, three of them claim there is no difference between women with and without endometriosis concerning 25(OH) vitamin D levels; however, the other six studies found significant differences regarding this aspect. (4) Conclusions: Our results underscored the complexity of analyzing the role of the vitamin D complex in a challenging condition like endometriosis and suggest that focusing on the tissue level might be essential to obtain accurate answers to our inquiries.

## 1. Introduction

Infertility and chronic pelvic pain, common but nonspecific symptoms, often indicate endometriosis, affecting 10–15% of women globally, with prevalence rising to 30–40% in infertility and approximately 33% in chronic pelvic pain cases [1,2,3]. The disease’s multifactorial nature involves a local immune-mediated inflammatory process and endocrine system imbalance [4,5]. Suggested mechanisms include retrograde menstruation, cellular metaplasia, and stem cell transfer through blood and lymphatics [6].

This prevalent gynecological condition, characterized by tissue resembling the uterine lining outside the uterus, is an estrogen-dependent chronic inflammatory process linked to hormonal imbalance [4,7]. Ectopic endometrial tissue can be found in various locations, such as the ovary, peritoneum, ligaments, bladder, intestine, or beyond the pelvic region [6].

Despite abundant data, understanding and treating endometriosis remains challenging. Hormonal treatments dominate, leaving women seeking pregnancy without suitable options [8]. Chronic inflammation leads to scar tissue, resulting in fibrosis and adhesions, manifesting in diverse symptoms from abdominal pain to psychological issues [6,9,10]. The complex relationship between endometriosis and infertility lacks a clear cause–effect link [10,11].

Studies have examined and continue to test several biomarkers in an effort to gain a better understanding of the endometriosis pathways and its therapy because there are still ongoing controversies in this area. Blood, urine, endometrial tissue, and peritoneal irrigation fluid were all used to screen for biomarkers. Due to their ease of collection, storage, and analysis, the evaluation of specific indicators in blood and urine is widely available for use [12,13].

Numerous studies from the literature have assessed the effects of medical interventions in an effort to illustrate expected improvements in endometriosis-related inflammation. A small portion of current research concentrates on vitamin D therapy for endometriosis [7,8,9,11,14]. Given its immunomodulatory, anti-inflammatory, antiproliferative, and anti-invasive properties, vitamin D has been suggested to play a part in the multifactorial pathogenesis of endometriosis [14]. Its blood levels affect the onset, development, and stage of endometriosis. Some studies have gone one step further and suggested that the polymorphism of vitamin D binding protein (VDBP) may operate as a risk factor for this illness [15,16,17]. Numerous biological effects of vitamin D are known to exist, but the modulation of the immune system is the one that has caught the attention of medical researchers. Studies using in vitro and animal models reveal the antiproliferative effect of vitamin D via triggering apoptosis [18,19,20].

This pleiotropic substance plays a variety of roles through the binding of 1,25 (OH)_2_ vitamin D to the vitamin D receptor (VDR), in addition to its pivotal role in calcium metabolism and bone mineralization. It notably impacts the endocrine, paracrine, and autocrine pathways through the transcription of certain genes. Vitamin D status controls estrogen-related pathways, which are connected to autoimmune illnesses [8,21,22]. It also affects T cell growth and function, macrophage activation, migration, and adhesion, cytokine generation, and the extracellular scavenger system. The liver converts vitamin D into 25(OH) vitamin D during the processing phase. Most of this metabolite’s circulation is VDBP-bound. Less than 1% of 25(OH) vitamin D is totally unbound, leaving just 10–15% of it bound to albumin. It is considered that the 25(OH) vitamin D fraction that is bound to albumin and free 25(OH) vitamin D are the bioavailable ones [23].

The role of vitamin D in the development and pathophysiology of endometriosis is still being determined. Endometriosis, one of the main pathologies investigated, results from an imbalance in the endometriotic cells. This disturbance causes them to become invasive, adhesive, and proliferate. Additionally, there is a link between endometriosis and a rise in the production of various inflammatory markers. Given that vitamin D is involved in a variety of reproductive system processes, evidence suggests that vitamin D production may also take place in endometriotic lesions, similar to that found in normal endometrium. The reproductive system also contains vitamin D receptors [24].

The gene that codes for VDBP is located on the long arm of chromosome 4 (4q11-q13), while the gene that codes for VDR is located on the long arm of chromosome 12 (12q12-q14). Even when a single nucleotide (SNP) is involved, the many polymorphisms of these two genes have the potential to modify gene expression with a variety of biological effects in cells that are selectively targeted. Many different systems, including the immune and reproductive ones, express the VDR gene. There are about 124 different VDBP variants. They are intimately related to the levels of vitamin D in the blood. Some of the VDR SNP polymorphisms, including FokI (F/f), BsmI (B/b), ApaI (A/a), and TaqI (T/t), were evaluated by Jafari et al. [25,26,27]. Their results indicated that there was no statistically significant distinction in the genotypes and allele frequencies of VDR and VDBP gene polymorphisms between individuals with endometriosis and those in the control group [25].

Conducting this review, we considered the recent surge in research interest aiming to underscore the role of vitamin D in the progression and severity of this perplexing ailment. We also performed a statistical analysis, a meta-analysis, of the results from the included studies, aiming to reveal the exact statistical impact of the obtained data. Our goal was to succinctly outline the implications and potential benefits of vitamin D therapy. The objective of this review is to scrutinize and summarize the correlation between vitamin D and endometriosis, shedding light on the role that vitamin D plays in the development and severity of this mysterious condition. First, we attempted to determine if there is a statistically proven relationship between vitamin D and endometriosis because of the conflicting data in the literature; second, we tried to determine if there is a relationship between the severity of endometriosis and vitamin D levels or if the discrepancies are related to regional sun exposure particularities.

## 2. Materials and Methods

According to the international Osteoporosis Foundation Guidelines, vitamin D levels were classified as follows: deficiency (<20 ng/mL), insufficiency (20–30 ng/mL), and sufficiency (≥30 ng/mL) [28]. All of the included studies used the same classification of vitamin D levels.

### 2.1. Search Strategy

ECM and IEB, two evaluators, independently reviewed the existing literature for significant studies published between January 2014 and January 2023. To reach a consensus, discrepancies were deliberated with a third reviewer (DRM). We employed all possible combinations of keywords and phrases to search the Cochrane Central Register of Controlled Trials (CENTRAL), Web of Science, and PubMed/Internet portal of the National Library of Medicine. The terms included “endometriosis”, “Vitamin D”, “25(OH)D”, “1,25(OH)D”, “25(OH)D3”, “1,25(OH)2D3”, “vitamin D binding protein”, and “vitamin D receptor”.

### 2.2. Dissemination and Ethics

Since this study is a review of the literature, ethical approval is not required.

### 2.3. Eligibility Criteria

*Inclusion criteria.* For inclusion in our review, eligible studies had to meet the following criteria: (1) inclusion of patients diagnosed with endometriosis through ultrasound or laparoscopic surgery, subsequently confirmed through histological examinations; (2) inclusion of case reports, observational studies, or clinical trials; (3) human studies; (4) studies comprising two groups: one with endometriosis and the other serving as a control; (5) investigations comparing vitamin D (ELISA) or VDBP, VDR, and VDBP levels between endometriosis groups and controls; and (6) articles published in any language.

The *exclusion criteria* comprised (1) women lacking confirmed endometriosis diagnosis; (2) studies lacking participant details; (3) animal and cell studies; (4) reviews, systematic reviews, and meta-analyses; (5) studies lacking a control group; (6) studies involving Vitamin D supplementation and hormonal treatment; and (7) studies presenting only abstracts without full-text articles.

### 2.4. Quality Assessment

Utilizing the Cochrane risk-of-bias tool for randomized studies, our two aforementioned investigators assessed the bias risk of the included studies. With the help of our third reviewer, we were able to resolve any discrepancies. 

### 2.5. Statistical Analysis

The results were computed using the free open-source Metafor R package (Version 4.4-0 2023, released on CRAN) [29] to conduct the statistical analysis of the data from the included studies.

The outcomes of the serum vitamin D level were determined using the random effect and fixed effect models. The Q statistic, the I2 measure, and the H2 measurements were calculated to examine the heterogeneity among investigations. Since 70% is the threshold number for I2, heterogeneity was the reason for a significant amount of variability among trials above this value. Since the REML (limited maximum likelihood) approach is known to produce unbiased estimates of variance components, it was chosen over the widely used DL (DerSimonian–Laird) method to assess this variability for the random model. Consequently, REML was used to approximate the τ (tau) variance. 

To visualize the effect, we used the Forest plot-type graph with a linear scale. Publications’ asymmetric variability was studied using a regression test for funnel plots.

Group 1 was defined as EMT (endometriosis) and group 2 was CON (control). Statistical measures like sample size, mean, standard deviation, effect (raw mean differences yi), variance (vi), significance (*p*), and confidence interval (ci) were revealed in tables for comparisons. 

Sample comparisons were also performed using a heterogeneity (variance) study, and a standard significance of 0.05 (5%) was used to define the cut-off for decisions on hypothesis testing.

## 3. Results

After utilizing the key terms to search across the databases, we were able to find 316 results. There were still 137 articles to screen once the duplicates were removed. After applying the exclusion criteria, we were left with only 15 publications from the selected period of time (2014–2022). Figure 1 depicts the flow diagram of the literature research procedure, and Table 1 below lists the resulting 15 publications.

Among these 15 papers, only nine articles were discovered to be complete or to have the ability to compute all the data from the ones that already existed for our study. We computed the missing data (average and standard deviation) using the values of the median and quartile. The nine studies are listed below in Table 2.

We attempted to determine whether the individual effect in these nine included studies has any statistically significant difference. Table 3 lists the measured effect using the raw average differences and accompanying variances.

The data from these nine included articles were computed first for the fixed-effects model and then for the random-effects model. Table 4 provides a description of the newly obtained indicators. Our statistical evaluation reveals that most of them are significant. Because of the significant degree of variability in the papers we considered, a fixed-effects model was not suited for our analysis.

Using a random-effects model we obtained a statistically significant *p* value < 0.001, with the following confidence intervals: lower bound 95% = −8.72, upper bound 95% = −2.42. When analyzing the variability between these studies, to evaluate if it impacts our results, as depicted in Figure 2 and Figure 3, we concluded that there can be other variables that provide variability.

The process of choosing publications should not be influenced by any biases. To assess whether there is any bias in the selection of publications, a funnel plot was created. This scatter plot displays the precision of the studies, which depends on their standard error. Ideally, the plot should appear symmetrical with respect to the vertical line representing the cumulative effect. To check for asymmetry, the Egger regression test can be employed. This test calculates a Z-type statistic and assesses its significance. In our analysis, we did not obtain a significant result (*p* = 0.36), indicating that no publication bias was detected.

The forest plot is utilized to summarize data from multiple studies and to visually represent the combined effect. Each study is represented by its effect size, confidence interval (represented by horizontal lines), and weight (depicted as black boxes). At the bottom of the plot, the combined effect (represented by a diamond shape) and its corresponding 95% confidence interval are displayed. As depicted on the chart, the effect is statistically significant, and the confidence interval is negative. This indicates that the effect signifies a decrease in vitamin D.

We totalized 976 patients with endometriosis and 674 controls. From these nine studies enrolled in our analysis, Lopes et al. focused mainly on infertility [30], and Hager in determining incidental endometriosis in women with clomiphene-resistant polycystic ovarian syndrome (CC-resistant PCOS) [34]. 

From January 1 through May 5, 2012, Lopes et al. acquired their data from a private assisted reproduction facility in Brazil, enrolling 232 infertile women and 137 controls [30]. They examined only 21 patients with endometriosis while primarily focusing on vitamin D levels and their implications in infertility. In this study, the diagnostic of endometriosis was established through video laparoscopy, and/or at least two vaginal ultrasounds 30 days apart. The authors do not mention any pathological confirmation of the diagnosis. The 25 women without endometriosis enrolled in the control group had a mean vitamin D level of 25.1 ± 8.5 ng/mL; meanwhile, the endometriosis group had a mean 25(OH) vitamin D level of 25.2 ± 7.9 ng/mL (*p* = 0.81), with no statistically significant differences between the studied group and the control one. In Lopes et al.’s cross-sectional retrospective study, for the control group, only data from 25 of the 137 recruited controls were used. The extents of endometriosis or the methodology utilized to select the control group were not disclosed in this study. The assessment for infertility was completed for only 14 of the 21 patients with endometriosis, and only three of them recognized endometriosis as an isolated pathology. Despite being a sunny region, 25(OH) vitamin D levels do not seem to suffer any improvement from the values of other regions that do not benefit from so much sunlight [30]. 

Hager et al.’s study is another one that appears to have a different objective than determining the 25(OH) vitamin D serum levels in endometriosis-affected women. The study provides no details about the probes’ collection, the time of year blood samples were taken, the subjects’ diets, vitamin D supplements, or medical care. They only examined accidental endometriosis diagnosis in a subset of patients with CC-resistant PCOS, specifically, in women who sought laparoscopic drilling. All instances with endometriosis that were identified through an ultrasound examination, as well as patients with endometriosis-suggestive symptoms, were excluded. The study was conducted in Vienna, Austria, between 2008 and 2018, over a ten-year span. With a 95% confidence interval (CI) of 0.982 (0.965; 0.999) in univariate analysis and a *p* value of 0.049, and with a 95% CI of 0.980 (0.962; 0.999) in multi-variate analysis with a statistically significant *p* value of 0.036, vitamin D levels were 14.0 ± 9.92 ng/mL in women with incidentally detected endometriosis and CC-resistant PCOS, compared to 17.52 ± 8.92 ng/mL in women with CC-resistant PCOS but without endometriosis. In their CC-resistant PCOS sample and their meta-analysis, the authors discovered a 16.9% and a 7.7% incidence of asymptomatic endometriosis, respectively. According to ASRM classification, the women who were found to have asymptomatic endometriosis were in stages I and II. According to this study, vitamin D levels were a good indicator of endometriosis [34].

Miyashita et al.’s cross-sectional study used the American Society for Reproductive Medicine’s (ASRM) classification for staging endometriosis. The study was conducted in Tokyo, Japan, and all the women enrolled, both in the study group and in the control one, were Japanese residents or Mongoloid. The serum was collected between October and March, during the fall and winter; moreover, because radioimmunoassay (RIA) proved to be less specific, sensitive, and accurate in evaluating 25(OH) vitamin D levels, the results were validated by using liquid chromatography associated with tandem mass spectrometry. The intriguing feature of this study is that it included both the study group and the controls after a laparoscopy result that either confirmed or ruled out endometriosis. The authors found no difference between 1,25(OH)_2_ vitamin D levels; however, the serum levels of 25(OH) vitamin D in patients with severe endometriosis (17.2 ± 1.1 ng/mL) were significantly lower than the levels detected in controls (21.8 ± 1.3 ng/mL, *p* < 0.05) and in patients with mild endometriosis (21.5 ± 1.4 ng/mL, *p* < 0.01). Nevertheless, the Japanese study did not solely focus on serum 25(OH) vitamin D levels; instead, the researchers also investigated the impact of 1,25(OH)_2_ vitamin D on human endometriotic stromal cells within an in vitro setting. An analysis of the various inflammation and proliferation markers in human endometriotic stromal cells demonstrated the biologically active form of vitamin D’s modulatory action [8]. 

In their 2017 letter to the editor, a case–control study by Anastasi et al. examined the levels of 25(OH) vitamin D in women with endometriosis and related pelvic pain. In the initial phase, 135 endometriosis patients and 90 controls were enrolled; however, only 104 patients with surgically and histologically proven endometriosis and 90 controls were kept after the case evaluation and application of the exclusion criteria. Women in the control group were those whose endometriosis had been clinically eliminated using ultrasonography and whose age and body mass index matched those of the study group. The women who were included were all Caucasians. Chemiluminescent enzyme immunoassay (CLEIA) was used to analyze the serum, which was collected between June and November 2015. In the endometriosis group, a detected serum level of 25(OH) vitamin D was 21.3 ± 8.9 ng/mL, with vitamin D deficiency or insufficiency present in 80% of cases. The values in the control group were statistically significant and higher (32.3 ± 2.67 ng/mL). With a *p*-value of 0.001, the scientists found a significant association between vitamin D deficit or insufficiency and endometriosis. Furthermore, any sort of pelvic pain was linked to low 25(OH) vitamin D levels [24].

Case–control research by Buggio et al. was another important study we considered in our analysis [31]. The 1990–2017 literature review on this subject is another source of this study’s strength. Six academic papers that compared patients with and without endometriosis for 25(OH) vitamin D levels were considered by the authors in their analysis. Anastasi et al. and Miyashita et al., two of our included research sources, were taken into account in their analysis as well [8,24]. Just five of the six included studies by Buggio et al. in their review examined actual 25(OH) vitamin D blood levels, and the sixth simply provided a prediction of vitamin D serum levels based on daily dietary intake [31]. Only Caucasian women with surgically or non-surgically diagnosed endometriosis were included in the Italian case–control research by Buggio et al. [31]. With the exception of iatrogenic post-C-section-determined endometriosis, the patients were classified into two groups (127 women with ovarian endometrioma and 90 women with deep invasive endometriosis). The recruiting phase lasted from October 2014 to January 2017. Rectovaginal plaques, bladder detrusor nodules, bowel lesions, intrinsic ureteral endometriosis, and deep endometriosis penetrating the pouch of Douglas and parametria comprised the deep invasive endometriosis cases included in their study. Women who visited the outpatient clinics for a variety of reasons were included in the control group. Endometriosis was excluded based on the patients’ anamnesis, clinical examination, and ultrasound pelvic evaluation. By excluding patients with vitamin D supplementation, including those with previous one-month, whole-body sun exposure, and by limiting the recruitment period to October through May, the authors aimed to reduce the impact of other external factors on 25(OH) vitamin D levels. The results were obtained using CLIA technology, with blinded biologists performing the serum analysis and evaluating the intra- and inter-assay coefficients of variation. In the endometriosis group, vitamin D serum levels were 17.9 (7.0) ng/mL and 18.4 (7.6) ng/mL in the control group (*p* = 0.46), with no statistically significant differences, not even between the group with ovarian endometrioma and deep infiltrating pelvic endometriosis. There were no statistically significant differences between women who had endometriosis and those who did not when the investigators divided 25(OH) vitamin D into its four subgroups (severe deficiency, deficiency, insufficiency, and sufficiency). This study is the first one encountered in our analysis that carefully examined the phenotypic traits of the women who were included, focusing on their current and previous sun exposure behaviors and their skin’s response to UV radiation, knowing that the majority of vitamin D reserves are produced by exposure to sunlight (90%) rather than dietary intake (10%). The fact that the authors deliberately chose their control group to closely resemble their study group is another component that strengthens this study [31]. 

Table 5 showed the research by Baek et al. is another Asian study that we examined. According to the ASRM classification, they have cases of both mild (stages I and II) and severe (stages III and IV) endometriosis that were discovered during laparoscopic procedures. The women who visited the gynecological department for a health checkup and did not report any endometriosis-related symptoms were chosen as the control group. The serum from these three groups was used to perform ECLIA to measure the total 25(OH) vitamin D level. The values for total vitamin D, VDBP, and serum albumin concentrations were used to compute the bioavailable and free 25(OH) vitamin D levels. With no differences between the three groups in terms of the levels of free and bioavailable 25(OH) vitamin D, their findings indicated a statistically non-significant negative correlation between 25(OH) vitamin D levels and the severity of the disease. However, advanced endometriosis was associated with significantly lower levels of the three vitamin D fractions compared to the healthy controls [32]. 

The outcomes of our analysis also considered a prospective Korean study by Cho et al. Comparative analyses of serum from 16 women with pathologically proven endometriosis and 16 healthy women who served as controls were performed. None of the 16 controls had any signs or symptoms of the condition. In addition to examining the levels of total, free, and bioavailable 25(OH) vitamin D, their investigation also looked at VDBP polymorphism. Serum samples were gathered between February 2017 and March 2019. The authors used an ECLIA technique to measure the serum levels of 25(OH) vitamin D [23]. As in the Baek et al. investigation, the amounts of free and bioavailable 25(OH) vitamin D were calculated. In contrast to the controls, women with endometriosis had statistically significant decreased levels of total 25(OH) vitamin D, according to Cho et al. Although the study group had lower levels of free and bioavailable 25(OH) vitamin D, the differences were not statistically significant. There was no endometriosis stage-based segregation of the study group in this study. According to a Korean epidemiological survey, over 80% of the women in that region are vitamin D deficient [23,32].

The case–control study conducted by Delbandi et al. in Iran from June 2017 to July 2018 also included Caucasian women. Laparoscopy was used in order to determine the diagnosis of endometriosis, which was then supported by histological analysis. The disease was staged by the authors using the ASRM classification. Women in the control group had benign conditions such as pelvic discomfort, uterine fibroids, unexplained infertility, as well as no history of endometriosis. High-pressure liquid chromatography (HPLC) was used to analyze the recruited women’s serum in order to detect 25(OH) vitamin D levels. In order to account for seasonal variation in 25(OH) vitamin D levels, the collection period was adjusted by ensuring that a similar number of patients would be assessed in each season. Only 8 of the 54 individuals included in the research group had endometriosis in stages I or II; the other 48 were in stages III or IV. In comparison to the control group, which had a serum 25(OH) vitamin D level of 22 ng/mL (16; 33 ng/mL), the levels of serum 25(OH) vitamin D were 8 ng/mL (5; 13 ng/mL) in stages I–II and 14 ng/mL (6; 30 ng/mL) in stages III–IV. In women with endometriosis, the authors found statistically significantly lower levels of 25 (OH) vitamin D; however, when they compared it by disease stage, they found no statistically significant higher amounts in stages III and IV [35].

The final study we used was one from 2021, carried out in Russia by Yarmolinskaya et al. The study included 30 women without gynecological pathology and 440 women with endometriosis that was laparoscopically identified and pathologically confirmed. The enzyme multiplied immunoassay (EMIT) was used to assess the levels of 25(OH) vitamin D. In comparison to controls, patients with endometriosis had median vitamin D levels that were lower than 22.1 (17.1; 28.0) ng/mL, compared to 36.0 ng/mL, respectively. Women with mild endometriosis had a median value of 23.6 ng/mL, whereas those in stages III–IV had a median value of 20.6 ng/mL; however, the difference did not achieve statistical significance. There is no information about the classification system used to stage endometriosis; moreover, there were no details about the season in which the serum preservations were performed, and no clinical characteristic details of the women included in the study were mentioned [36].

When we attempted to compare research from different climate zones that were similar in terms of solar exposure among each other, we were unable to find any statistically significant findings. However, when contrasting the findings of studies conducted in more sunny regions with those conducted in less sunny regions, statistically significant differences were found with a *p* value of 0.004 (Yarmolinskaya et al., Delbandi et al., and Hager et al. compared with Miyashita et al. and Lopes et al.) [8,30,34,35,36].

From the nine studies included in our analysis [8,23,24,30,31,32,34,35,36], three of them claim there is no difference between women with and without endometriosis concerning 25(OH) vitamin D levels (Lopes et al., Buggio et al., and Cho et al.) [23,30,31]. Significant differences in 25(OH) vitamin D levels, between endometriosis-affected women and controls, were found in all six studies [8,24,32,34,35]. Furthermore, there were noticeable disparities in Miyashita et al., even across women with different stages of the disease, with vitamin D status being indirectly correlated with the endometriosis stage [8]. Delbandi et al.’s study, which demonstrated higher 25(OH) vitamin D levels in women with advanced stages of the disease compared with the first two stages of endometriosis, somewhat refuted this finding; however, the difference did not reach statistically significant levels [35].

We attempted to divide the results of the included studies into cases with mild (stages I and II) and severe (stages III and IV) endometriosis, according to the ASRM classification, and compare serum vitamin D levels because the results from our nine included studies were inconsistent concerning the correlation between serum vitamin D levels and the endometriosis stage. We examined serum levels of vitamin D using a random effect model. Although there was a clear difference in the point values, 13.9319 in severe endometriosis versus 19.6533 in mild endometriosis, the confidence intervals overlapped because of the larger variance; therefore, there was no statistically significant difference (*p* value of 0.3246 by Q heterogeneity statistic).

Only Haeger et al.’s data, which were in nmol/L, required conversion to allow comparison with other study results; the rest reported their outcomes in ng/mL [34].

## 4. Discussion

Contrary to the initial assumption that the bioavailable fractions could be more therapeutically relevant, the level of 25(OH) vitamin D appears to be the most effective approach for determining vitamin D status. Many in vivo and in vitro findings support the fact that vitamin D supplementation might play a beneficial role in endometriosis regression. Vitamin D treatment has a proven therapeutic impact on endometriotic implants, according to animal research. By performing endometrial auto-transplantation to the peritoneum in mice models, surgically inducing endometriosis, Abbas et al. and Yildirim et al. demonstrated that the implants regressed after the intraperitoneal or intramuscular injection of vitamin D [38,39]. The research of Mariani et al. supports these conclusions as well. Moreover, oral vitamin D supplementation proves to have the same effect in mice with endometriosis, a disease determined through the injection of endometrial tissue obtained from donor mice [40]. The outcomes of these three earlier investigations are in disagreement with a more current animal study by Akyol et al. Regarding the demise of endometriosis implants following intraperitoneal vitamin D injection, the investigators found no improvement [41]. The in vitro studies included in Kalaitzopoulos et al.’s systematic review support the reduction in inflammation after vitamin D administration [42].

As presented further in human studies, there are many inconsistencies in the results. Endometriosis is a very heterogeneous disease. It has four stages of severity according to ASRM classification and many phenotypes (peritoneal, ovarian, and deep-infiltrating endometriosis). These aspects might contribute to the discrepancies that manifest themselves in the literature. Of our nine included studies [8,23,24,30,31,32,34,35,36], Miyashita et al., Baek et al., and Yarmolinskaya et al. used ASRM classification; however, only two of them provided the number of patients from each stage that were included in the study [8,32,36]. In Lopes et al.’s study, there is no information about the stage of the disease, and all the women included with endometriosis suffered from infertility [30]. In Hager et al.’s study about the incidentally found endometriosis in women with CC-resistant PCOS, we believe that all the patients diagnosed with endometriosis were in stage I of the disease, especially due to the lack of any symptoms and complaints [34]. Miyashita et al. included in their study 17 women in stages I and II of the disease and 22 patients in stages III and IV according to the ASRM classification [8]. The subgroup of women from the Anastasi et al. and Cho et al. studies were not divided at all depending on the endometriosis stage of the disease; the first aforementioned study focused more on the symptoms, evaluating 25(OH) vitamin D levels in women with endometriosis and associated pelvic pain [23,24]. Buggio et al. included in their study women with ovarian endometrioma and with deep lesions, meaning that all the patients were in stages III or IV of the disease [31]. Baek et al. and Delbandi et al. classified the patients using ASRM into mild endometriosis for the first two stages and advanced endometriosis for the last two stages; however, the number of patients included from each group was not equitable as in Baeks’s study. Delbandi included eight patients in the first two stages and 46 in the last two [32,35]. Yarmolinskaya et al.’s study was the largest if we analyze the number of patients, even if they used the ASRM classification to divide women with endometriosis into two subgroups (mild and prevalent endometriosis); there is no information about the exact number of patients from each of the subgroup included in their analysis, with the research focusing mainly on the treatment [36]. When we separately analyzed the difference between vitamin D levels in mild and severe endometriosis, our results revealed no statistically significant difference.

Excluding the studies by Lopes et al. and Hager et al. that did not concentrate on 25(OH) vitamin D in endometriosis-affected women, our findings showed that there are significant differences between serum 25(OH) vitamin D levels in women with endometriosis depending on sun exposure but no significant differences between the mild and severe stages of this disease [30,34].

The authors’ comparative control groups may have also contributed to discrepancies in their findings about 25(OH) vitamin D levels. Women in the control group of Lopes et al.’s study were between the ages of 30 and 45 and had no complaints of infertility [30]. The authors examined the 25(OH) vitamin D serum levels of 21 women with endometriosis who had undergone video-laparoscopic diagnosis with those of 25 controls and also with women who had other causes of infertility. They discovered no differences in these three categories [30]. Low serum 25(OH) vitamin D concentrations have been connected to lower reproductive rates; however, this connection has not always been supported by the literature [43]. This element might be a useful justification for why Lopes et al.’s comparison of the infertile groups failed to reveal any differences. We were able to explain half of these results by assuming that lower vitamin D serum levels are related to infertility. Only 3 of the 14 women with surgically confirmed endometriosis who underwent a complete couple’s infertility evaluation had this one infertility factor alone; the other 11 had endometriosis in combination with other infertility factors that could have an impact on the serum results. The authors offer no other information outside the two facts listed above about the control group’s age distribution and the absence of any complaints regarding women’s fertility [30]. Miyashita et al.’s research established the selection criteria in great detail. They enlisted patients with moderate endometriosis, which was proven using laparoscopy (stages I and II according to the ASRM classification), severe endometriosis, which was confirmed through surgery (stages III and IV), and a total of 37 age-matched controls with surgically infirmed endometriosis. Along with accurately classifying the patients into these three groups, the authors also considered the method’s reduced specificity and sensitivity for detecting 25(OH) vitamin D, and they confirmed the RIA results using tandem mass spectrometry and liquid chromatography [8]. In order to reduce bias, they collected serum from patients in the fall and winter, two seasons that are comparable in terms of sunlight exposure. They also excluded women who had fibroids from the analysis and collected the serum during the menstrual cycle’s proliferative phase, based on the presumption that vitamin D levels vary with different menstrual cycle phases [44]. Following our included publications, Anastasi et al. investigated vitamin D levels during the proliferative phase of the menstrual cycle to prevent any alleged, but still not yet universally proved, changes in vitamin D levels during the menstrual cycle [24,44]. The control group’s age and body mass index (BMI) matched those of the women from the study group. The authors chose a control group of 90 women who had no endometriosis symptoms, clinical signs, or ultrasound findings. The same criteria used by Anastasi et al. were also utilized by Buggio et al., who premised that incidental endometriosis without any clinical symptoms is too uncommon to be a source of error [24,31]. But, as with Cho et al. later in their study, Buggio et al. excluded from both the study group and control group women with malignancy, uterine leiomyomas, hypertension, diabetes, multiple sclerosis, autoimmune disorders, and coronary, hepatic, or renal diseases [23,31]. Baek et al., Cho et al., Yarmolinskaya et al., and Delbandi et al. based their control group selection solely on the absence of endometriosis symptoms during routine gynecological screening [23,32,35,36]; however, only Cho et al. used laboratory tests or questionnaires to exclude women with uterine leiomyomas, hypertension, diabetes, as well as multiple sclerosis, auto-immune disorders, or coronary, hepatic, or renal disease to prevent any influence on 25(OH) vitamin D levels [23]. The only authors who included women with benign gynecologic complaints were Delbandi et al. [35]. While excluding all cases of malignancies, adenomyosis, endometrial cancer, hyperplasia, endometrial polyps, autoimmune disorders, cardiovascular disease, any other acute or chronic inflammatory disorders, infectious diseases, pregnancy, smoking, alcohol consumption, hormone therapy, and vitamin D supplementation or medication that influences bone metabolism, they included women with unexplained infertility, uterine fibroids, and pelvic pain, with an absent history of endometriosis [35]. Despite the broad exclusion criteria, this study’s results can be questioned because of the allegedly common correlation between unexplained infertility and pelvic pain with inadequate vitamin D levels [35]. Even though the incidence of asymptomatic endometriosis is low in the general population, somewhere between 3 and 7.7%, and its impact on study statistics could be assumed to be very modest, we still need to consider it when choosing the control group [44,45,46]. Delbandi et al.’s research considered the hypothesized change in 25(OH) vitamin D serum levels dependent on the menstrual cycle phase [35] in addition to Miyashita et al. and Anastasi et al. Moreover, Delbandi et al. assessed the menstrual cycle phases for each participant, whether they were in the study group or the control group. In the control group, there were more women in the proliferative phase, while there were more women in the secretory phase in the endometriosis group [8,24,35]. This factor represents yet another potential source of inaccurate outcomes. Yarmolinskaya et al. compared the serum 25(OH) vitamin D levels of 440 women with laparoscopically diagnosed, pathologically proven endometriosis and endometriosis pain-related symptoms with those of only 30 women in their control group. The number of women included in the endometriosis study group and the control group is quite out of balance, posing serious concerns about the accuracy of the results obtained [36]. Hager exclusively included women who underwent laparoscopic ovarian drilling and met the Rotterdam criteria for CC-resistant PCOS, excluding all the women with clinical and para-clinical findings that might suggest endometriosis [34]. As is well known, women with PCOS have several unique characteristics compared to those without, and PCOS itself affects vitamin D blood levels. These two elements can be a source of inaccurate results [47].

We carefully analyzed the three studies that found no statistically significant difference between women with and without endometriosis, which concerns 25(OH) vitamin D levels, namely, Lopes et al., Buggio et al., and Cho et al. [23,30,31]. The initial study solely examined the relationship between vitamin D levels and infertility, examining a limited sample of endometriosis-positive women and contrasting them with other infertile women who had levels of vitamin D that were not necessarily a reflection of the general population levels or related to their infertility. The value of their results, which concern women with infertility, is undeniable; however, those obtained in women with endometriosis can be questioned [23]. Buggio et al.’s study included 217 women with advanced-stage endometriosis (ovarian endometriotic cyst and deep-infiltrating endometriosis), comparing their 25(OH) vitamin D serum levels to those of 217 healthy controls. Although the study benefits from a large number of women, including all women in the study group with endometriosis stage III or IV, the authors did not exclude patients undergoing hormonal therapies. More than half of the women included in the study group (128 women)—30% of those in the control group—were on estroprogestins, progestins, and GnRH analogs [31]. Although there is debate in the literature regarding the way in which hormonal therapy influences 25(OH) vitamin D levels, with many studies reporting an increase in serum vitamin D levels in women under such treatment, a Turkish study by Namli Kalem et al. states the contrary, despite the extremely valuable work of Buggio et al.; nevertheless, we still cannot consider their findings as unquestionable [31,48]. The last of our examined studies, the one accomplished by Cho et al., detected a negative correlation between 25(OH) vitamin D levels in women with endometriosis compared with controls; however, bioavailable and free 25(OH)D levels were similar between the endometriosis and control groups (*p* = 0.858 and *p* = 0.961, respectively) [23]. Given the short and active half-life form, we cannot exclude a correlation based on only this aspect [20].

Moreover, Baek et al. detected in their study a negative correlation between the endometriosis stage of the disease and vitamin D; however, the difference was not statistically significant. This might be attributed to the small sample size, and perhaps a larger study will confirm this indirect association [32].

After eliminating studies that concentrated on other pathologies, did not classify endometriosis according to ASRM, had only severe stage endometriosis, and had inequitable differences between the study group and the control group, as well as stringent criteria for selecting the control group, we only had one study left to analyze, the one performed by Miyashita et al. [8]. The serum levels of total, bioavailable, and free 25(OH)D were significantly lower in the severe endometriosis group than in the healthy control group (*p* = 0.001, *p* = 0.018, and *p* = 0.049, respectively), despite the authors’ finding that there was a negative correlation between 25(OH) vitamin D levels and the endometriosis stage. However, the difference did not reach statistical significance. Being a study with a small number of included cases, extensive additional research is needed to confirm their findings, applying the same strictness in selecting the included cases [8].

Another source of conflicting results might reside in the fact that the gold standard method to assess 25(OH) vitamin D levels is liquid chromatography–tandem mass spectrometry (LCMS/MS). But, as stated by Knudsen et al., by using an electrochemiluminescent assay (ECLIA), one can obtain comparable results. Only a single study from our included selection used the gold-standard method of detection, namely, the one conducted by Miyashita et al.; all eight other studies used CLIA, ECLIA, or HPLC [49].

To be able to compare these nine studies’ results, we need to carefully analyze the regions in which they were conducted, the vitamin D status regional particularities, and the time frame for serum collection [8,23,24,30,31,32,34,35,36]. Lopes et al.’s study was conducted in Brazil, a country with equatorial, tropical, as well as sub-tropical climates. Being located in the center of Brazil, Brasilia has an equatorial climate [30]. In Brazil, the prevalence of vitamin D deficiency or insufficiency was 15.8% among white women and 25% among brown and black women according to Santana et al., which is similar to Lima-Costa et al.’s 2020 ELSI study [50,51]. In Lopes et al.’s research, 32.9% of the women with infertility met the criteria for vitamin D supplementation. They used the chemiluminescent immunoassay (CLIA) technique to measure 25(OH) vitamin D levels [30]. The study by Miyashita et al. was carried out in Tokyo, Japan, which has a subtropical climate [8]. The prevalence of vitamin D deficiency/insufficiency is considered to be 47.7% in summer and 82.2% in winter [52]. Serum evaluation for vitamin D status was performed using liquid chromatography–tandem mass spectrometry. Our third and fourth investigations were conducted in Italy, another country with a warm, Mediterranean environment that experiences an abundance of sunlight. According to a 2021 study by Morsilli et al., young women only acquire adequate vitamin D levels near the end of the summer, and hypovitaminosis D is a common problem in women, even in rural locations with more solar exposure [53]. In their research, 22% of women had insufficient serum levels, and 68.6% were vitamin D deficient. The examination was carried out using a chemiluminescent microparticle immunoassay, and the mean blood concentrations of 25(OH) vitamin D in women were 17.2 ± 10.0 ng/mL [53]. To eliminate seasonal fluctuation, the women who had been recruited in the Anastasi et al. study were invited back in July to collect serum for 25(OH) vitamin D measurements [24]. The next two scientific works included took place in South Korea, with a humid subtropical climate. The median 25(OH) vitamin D levels in women were 20.5 ng/mL, with vitamin D deficiency prevalence being higher in young women. Vitamin D serum levels were assessed using chemiluminescence microparticle immunoassay [24,54]. We searched for vitamin D status information in Austria, a nation with a temperate environment, where the Hager et al. study was conducted [34]; however, we could only locate a 2017 study by Eldmafa et al. Vienna, which is located in the eastern part of Austria, experiences a continental climate [55]. High-performance liquid chromatography with ultraviolet (HPLC with UV) was used in this investigation to determine the levels of vitamin D. Women between the ages of 18 and 24 had 25(OH) vitamin D levels of 21.8 ng/mL while women between the ages of 24 and 50 had values of 21.56 ng/mL, with 30.2% of women having values below 12 mg/dL [55]. The prevalence of vitamin D deficiency in women was found to be 61.90% in an Iranian review and meta-analysis published in 2018; nevertheless, there were significant regional differences. However, compared to men, women have a higher frequency of vitamin D insufficiency, according to scientists. The prevalence of vitamin D insufficiency is very high in Iran due to its continental climate [56]. Another area with a humid continental climate is Saint Petersburg, where the final study we included was conducted. Vitamin D deficiency was found in 50.4% of Russian women, whereas vitamin D insufficiency was found in 33.7% of them, according to Karonova et al.’s study, which used electrochemiluminescence immunoassay to perform the analysis. Therefore, it is exceedingly challenging to compare the findings of studies from such climatically diverse places [57]. Our findings showed there are statistically significant differences between serum vitamin D levels from warmer regions and regions that do not benefit from much sunlight exposure, taking into consideration that the skin is where most vitamin D production takes place, requiring UV solar radiation.

The study conducted by Buggio et al. obtained similar results to those of Agic et al., stating that there was no difference between 25(OH) vitamin D serum levels in women with endometriosis and in those without [31,58]. Agic et al. limited the time period for serum collection between April and September and chose their research and control group based on endometriosis laparoscopic findings [58]. Both of the studies used ASRM classification to stage endometriosis [31,57], but Buggio et al. only included women in stages III or IV of the disease [31]. The results were obtained using a radioimmunoassay to measure the levels of 25(OH) vitamin D; however, as was previously mentioned, RIA is a potential source of error in this instance [31]. Agic et al.’s findings may also be called into doubt because the study did not take into consideration some of the main factors that seriously influence serum 25(OH) vitamin D levels, such as BMI, age, and dietary intake, and also because they date back to 2007, and more advanced and accurate methods of vitamin D measurement have since been developed [58].

Additionally, due to its brief half-life, 1,25(OH)_2_ vitamin D serum levels in patients cannot be studied with greater accuracy. One cannot dispute the association between vitamin D levels and endometriosis in light of these findings.

The study design, sample size, and the difficult methodological evaluation of vitamin D’s effect on endometriosis are all factors that may be associated with the risk of biased assessment, which may explain the variations in study results. It is still possible that endometriosis-related inflammation is what causes reduced vitamin D levels. In addition, a wide range of variables, including smoking, diet, obesity, and, of course, sun exposure, affect vitamin D levels, affecting the results and making it hard for us to compare and draw some valid conclusions. Another aspect that needs to be taken into consideration is VDR polymorphisms that have a key role in its bioavailability. There are not many studies in the literature that have analyzed this aspect, together with 25(OH) vitamin D serum levels. Some of the studies included in our analysis have a gradient present, revealing differences between patients with advanced stages of the disease and those with mild ones, underlining the importance of disease staging when assessing vitamin D levels. Moreover, the risk of publication bias needs to be considered.

## 5. Conclusions

Our discussions emphasized the complexity inherent in analyzing the role of the vitamin D complex in a challenging condition like endometriosis, given the numerous variables to consider. It suggests that directing our attention at the tissue level may be crucial in obtaining accurate answers to our questions.

## Figures and Tables

**Figure 1 jpm-14-00119-f001:**
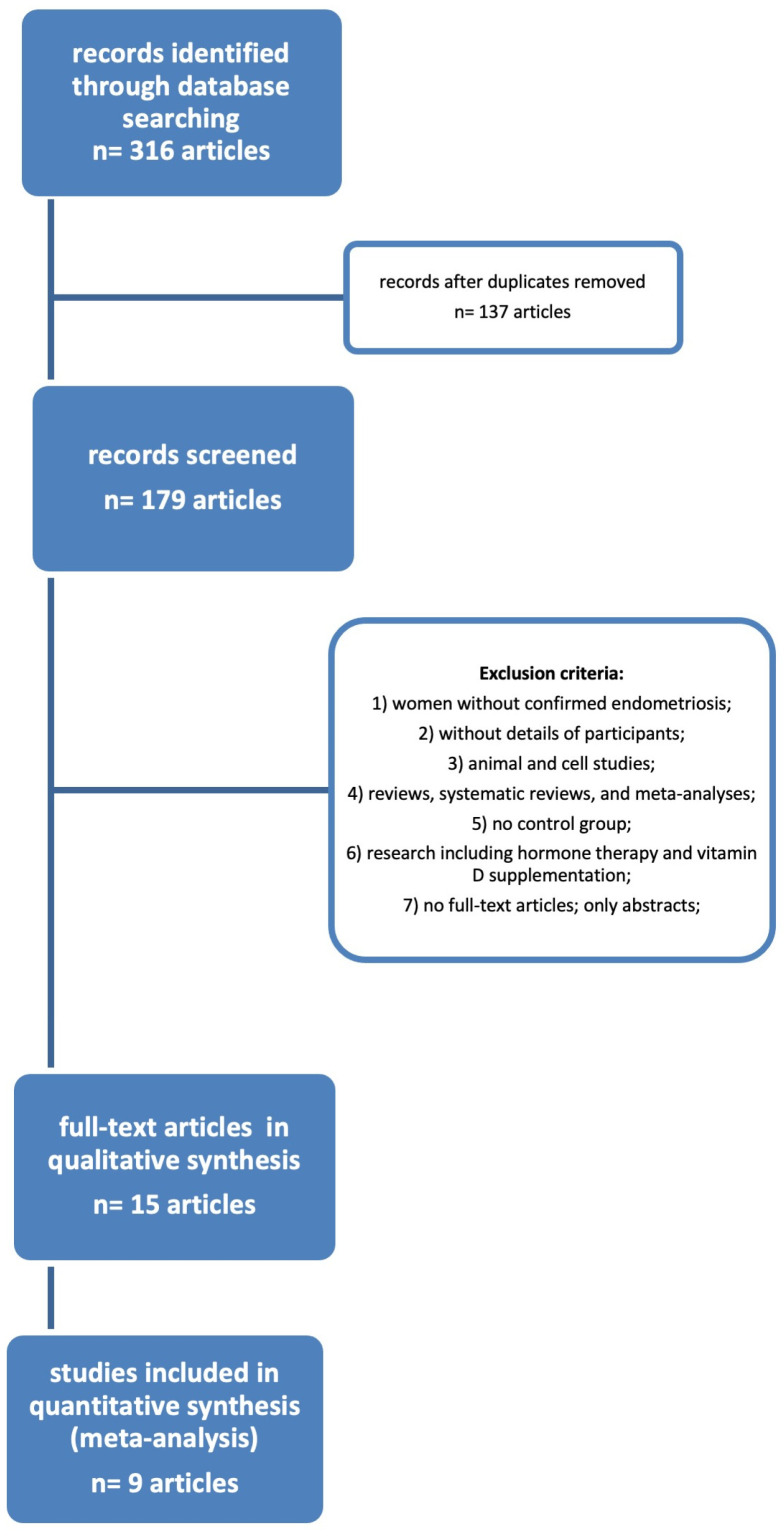
Flow diagram of the search process.

**Figure 2 jpm-14-00119-f002:**
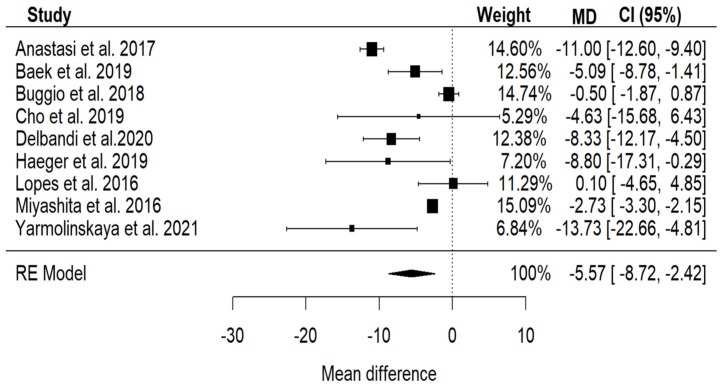
Forest plot for our nine included studies [8,23,24,30,31,32,34,35,36].

**Figure 3 jpm-14-00119-f003:**
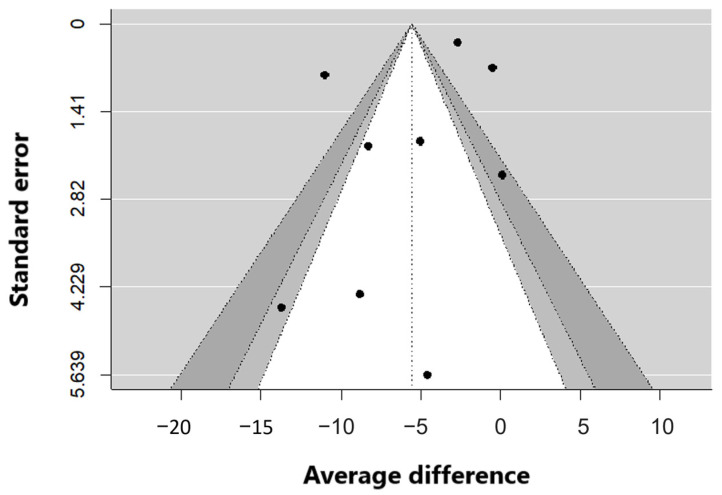
Publication error–random effect model.

**Table 1 jpm-14-00119-t001:** The 15 included articles with their statistical results.

Report (Author, Years of Publication)	Country	Design of Study	No. of Cases		Results						Key Findings
			EM	C	Tissue	outcome	Endometriosis		Control		
Szczepanska et al., 2015 [11]	Poland	-	154	347	serum	genotyping VDBP (PCR) genotyping VDR	SNPs (rs7041) polymorphisms gene (FokI, BsmI,)				The distribution of the VDBP and VDR genotypes was not significant between patients and control group (*p* > 0.05)
Lopes et al., 2016 [30]	Brazilia	RCSS	21	21	serum	25(OH)D—ELISA	25.2 ± 7.9 ng/mL		25.1 ± 8.5 ng/mL		No significant difference between groups (*p* = 0.81)
Miyashita et al., 2016 [8]	Japan		39 grade I–II: n = 17 grade III–IV: n = 22	37	serum	25(OH)D	21.5 ± 1.4 ng/mL 17.2 ± 1.1 ng/mL		21.8 ± 1.3 ng/mL		Significantly lower in severe endometriosis and CON (*p* < 0.05) and mild endometriosis (*p* < 0.01)
Derakhsshan et al., 2018 [7]	Iran	RCS	200	154	serum	25(OH)D ELISA	80.9% < 30 ng/mL		58.4% < 30 ng/mL		25(OH)D deficiency was significantly higher in the EMT group compared to CON (*p* < 0.001, [OR] = 29.4
Buggio et al., 2018 [31]	Italy	CCS	217 endometrioma n = 127 deep: n = 90	217	serum	25(OH)D ELISA	17.9 (7.0) ng/mL: 17.3 (7.4) ng/mL 18.7 (7.4) ng/mL		18.4 (7.6) ng/mL		Association between serum 25(OH)D levels and EMT was not significant
Baek et al., 2019 [32]	Korea	OS	9 mild EMT 7 advanced EMT	16	serum	25(OH)D - ELISA VDBP- ELISA	14.17 ± 7.62 ng/mL 8.91 ± 1.67 ng/mL 161.25 ± 53.09 ng/mL 198.34 ± 42.54 ng/mL		16.96 ± 4.71 ng/mL 169.20 ± 36.31 ng/mL		For serum 25(OH) and VDBP; no significant differences between the control group and EMT. 25(OH)D had a negative relationship with the severity of endometriosis
Lee et al., 2019 [33]	Korea	CCS	30	30	tissue	VDR polymorphisms (PCR sequencing)	polymorphisms (FokI, BsmI, ApaI, and TaqI)				The associations between the four polymorphisms of the VDR gene and endometriosis were not significant
Cho et al., 2019 [23]	Korea	PS	16	16	serum	25(OH)D—ELISA VDBP—ELISA	9.55 ng/mL 173.06 ng/mL		16.48 ng/mL 158.58 ng/mL		The total serum 25(OH)D level was significantly lower in the EMT than CON (*p* = 0.017); VDBP had no significant differences between the control group and EMT (*p* = 0.323)
						genotyping VDBP (PCR)	SNPs (rs4588 and rs7041)				The genotype and allele frequencies were similar between the two groups (*p* = 0.788 and *p* = 0.946)
Haeger et al., 2019 [34]	Austria	RCS	38	187	serum	25(OH)D ELISA	35.0 ± 24.8 nmol/L (14.02 ± 9.93 ng/mL)		43.8 ± 22.3 nmol/L (17.54 ± 8.93 ng/mL)		Serum 25(OH)D levels were significant (*p* = 0.049)
Delbandi et al., 2020 [35]	Iran	CCS	54	56	serum peritoneal fluid (PF)	25(OH)D—ELISA	12 ng/mL 4 ng/mL		22 ng/mL 6 ng/mL		Serum and PF levels of 25(OH)D in the EMT group were significantly different from that in the control group (*p* = 0.001 and *p* = 0.03)
Yarmolinskaya et al., 2021 [36]	Russia	-	440: grade I–II grade III–IV	30	serum (n = 440) peritoneal fluid (PF) (n = 49)	25(OH)D ELISA	I–II: 23.6 ng/mL III–IV: 20.6 I–II: 8.9 ng/mL III–IV: 3.6 ng/mL		36.0 ng/mL		25(OH)D levels in PB were not significant but in PF was significantly dependent (*p* = 0.004)
			129	82		VDR polymorphisms genes (PCR)	polymorphisms BsmI				Genotype of VDR BsmI gene polymorphic increased endometriosis risk
			27	20	tissue—endometrium	VDR expression level (IHC)	proliferative phase	secretory phase	proliferative phase	secretory phase	
							20.12%	21.51%	14.76%	22.35%	
Jafari et al., 2021 [25]	Iran	CCS	120	110	serum	VDR genotyping (DNA) VDBP genotyping	polymorphisms gene (FokI, BsmI, ApaI, and TaqI) polymorphic alleles (GC*1S, GC*1F, GC*2)				VDR and VDBP gene polymorphisms had no significant associations with endometriosis
De Pascali et al., 2021 [37]	Italy	-	8	37	tissue—endometrium	VDR monoclonal (IHC)	IRS in the nuclei of glandular ovarian endometrial cells				No significant expressions of cytoplasmic VDR (*p* = 0.053)
Chen et al., 2022 [12]	Taiwan	-	8	19	urinary	VDBP ELISA	90.657 ng/mL		51.385 ng/mL		Not significant (*p* = 0.095)

**Table 2 jpm-14-00119-t002:** The nine included articles with either complete or computable statistics and their study results.

Study_Name	n_EMT	n_CON	Mean1	Mean2	Std1	Std2
Anastasi et al., 2017 [24]	135	90	21.3	32.3	8.9	2.67
Baek et al., 2019 [32]	16	16	11.868	16.96	5.863	4.71
Buggio et al., 2018 [31]	217	217	17.9	18.4	7	7.6
Cho et al., 2019 [23]	16	16	12.32	16.947	17.409	14.344
Delbandi et al., 2020 [35]	54	56	15.333	23.666	6.76	12.9356
Haeger et al., 2019 [34]	38	187	35	43.8	24.8	22.3
Lopes et al., 2016 [30]	21	25	25.2	25.1	7.9	8.5
Miyashita et al., 2016 [8]	39	37	19.074	21.8	1.238	1.3
Yarmolinskaya et al., 2021 [36]	440	30	22.4	36.133	8.107	24.85

**Table 3 jpm-14-00119-t003:** The cumulative effect of the nine included studies.

No.	Study_Name	Yi	Vi
1.	Anastasi et al., 2017 [24]	−11	0.66595074
2.	Baek et al., 2019 [32]	−5.092	3.53492931
3.	Buggio et al., 2018 [31]	−0.5	0.49198157
4.	Cho et al., 2019 [23]	−4.627	31.80147606
5.	Delbandi et al., 2020 [35]	−8.333	3.83428305
6.	Haeger et al., 2019 [34]	−8.8	18.84456797
7.	Lopes et al., 2016 [30]	0.1	5.86190476
8.	Miyashita et al., 2016 [8]	−2.726	0.08497424
9.	Yarmolinskaya et al., 2021 [36]	−13.733	20.7334548

**Table 4 jpm-14-00119-t004:** Random effect model evaluation of the nine included studies.

No.	Study_Name	Yi	Vi	Sei	Zi	Pval	Ci.lb	Ci.ub
1.	Anastasi et al., 2017 [24]	−11	0.666	0.8161	−13.4794	<0.0001	−12.5994	−9.4006
2.	Baek et al., 2019 [32]	−5.092	3.5349	1.8801	−2.7083	0.0068	−8.777	−1.407
3.	Buggio et al., 2018 [31]	−0.5	0.492	0.7014	−0.7128	0.4759	−1.8747	0.8747
4.	Cho et al., 2019 [23]	−4.627	31.8015	5.6393	−0.8205	0.4119	−15.6798	6.4258
5.	Delbandi et al., 2020 [35]	−8.333	3.8343	1.9581	−4.2556	<0.0001	−12.1709	−4.4951
6.	Haeger et al., 2019 [34]	−8.8	18.8446	4.341	−2.0272	0.0426	−17.3083	−0.2917
7.	Lopes et al., 2016 [30]	0.1	5.8619	2.4211	0.0413	0.9671	−4.6453	4.8453
8.	Miyashita et al., 2016 [8]	−2.726	0.085	0.2915	−9.3515	<0.0001	−3.2973	−2.1547
9.	Yarmolinskaya et al., 2021 [36]	−13.733	20.7335	4.5534	−3.016	0.0026	−22.6575	−4.8085

**Table 5 jpm-14-00119-t005:** Vitamin D serum level differences between mild and severe endometriosis.

No.	Study_Name	n_Size	Mean_d	Std_d	Group_s
1	Baek et al., 2019 [32]—mild	9	14.7	7.62	2
2	Baek et al., 2019 [32]—severe	7	8.91	1.67	1
3	Buggio et al., 2018 [31]—severe	217	17.9	7	1
4	Cho et al., 2019 [23]—severe	16	9.55	2.674	1
5	Delbandi et al., 2020 [35]—mild	8	8.67	5.93	2
6	Delbandi et al., 2020 [35]—severe	46	16.67	17.78	1
7	Haeger et al., 2019 [34]—mild	38	35	24.8	2
8	Miyashita et al., 2016 [8]—mild	17	21.5	1.4	2
9	Miyashita et al., 2016 [8]—severe	22	17.2	1.1	1

## Data Availability

The data used to support the findings of this study are available upon request to the corresponding author.

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
