# Peer review of "Vitamin D—The Iceberg in Endometriosis—Review and Meta-Analysis"

_jpm, 2024, doi:10.3390/jpm14010119_

Round 1

Reviewer 1 Report

Comments and Suggestions for Authors

The Report ‘’ Vitamin D- the iceberg in endometriosis ’’ was written by Alexandra Urasche et al.

In this article, the literature on the relationship of vitamin D with endometriosis, which has been little researched before, is examined. The literature review is based on researching the Cochrane Central Register of Controlled Trials (CENTRAL), Web of Science and Pubmed/Internet portal of the National Library of Medicine databases using keywords related to the subject. As a result of the article, it was seen that vitamin D levels showed significant differences depending on sun exposure, but no significant difference was observed between mild and severe stages of endometriosis. The paper also suggests that this highlights the complexity of analyzing the role of vitamin D and that it may be necessary to focus on the tissue level to obtain accurate answers.

Endometriosis is an important cause of infertility and pelvic pain symptoms, which are increasingly common today. Many mechanisms have been put forward to elucidate the emergence of the disease and many factors that trigger the disease have been suggested.

­­

Vitamin D is a vitamin whose relationship with endometriosis has not yet been clearly established and needs to be investigated. Vitamin D plays an important role in autoimmune diseases, immunity and many physiological events. Revealing its clear relationship with ednometriosis may provide enlightening data both during the treatment process and the patient's presentation.

Grammar Correction;

·      There is a lack of meaning in the sentence in the introduction, “Studies using in vitro and animal models continue to have an anti-proliferative effect via triggering apoptosis [18-20]. ‘’ , it should be stated that this effect has been shown to be dependent on vitamin D.

·      There is a grammatical error in the sentence "Less than 1% of 25(OH) vitamin D is totally unbound, leaving just 10%– 89 15% of it bound to albumin." in the Introduction section. It should be as follows: "Less than 1% of 25(OH) vitamin D is totally unbound, leaving just 10%– 89 15% of it bounds to albumin."

·      In order to make the sentence "The reproductive system also contains its receptor" in the introduction more understandable, it should be stated that the receptor is the receptor of vitamin D.

·      In the Introduction section, ''Some 106 of the VDR SNP polymorphisms, including FokI (F/f), BsmI (B/b), ApaI (A/a), and TaqI (T/t), were evaluated by Jafari et al [25 -27].'' sentence, the study was mentioned, but not providing information about the data obtained as a result of the study created a deficiency.

·      There is a grammatical error and lack of meaning in the sentence ‘’The Japanese study, however, looked at more than just serum 25(OH) vitamin D levels; the scientists also examined the effects of 1,25(OH)2 vitamin D on human endometriotic stromal cells in an in vitro context. ‘’  in the results section.

·      To enhance the readers’ attention I strongly recommend referring these articles which were published recently.

·      I would like to re-review after revisions.

·      Sincerely,

Comments on the Quality of English Language

See the above

Author Response

Dear reviewer,

Thank you for your efforts in reviewing our work and for your comments and your suggestions that aimed to improve it. Your comments have provided valuable insights that we believe will contribute to enhancing the overall quality of our work.

Grammar Correction;

  • There is a lack of meaning in the sentence in the introduction, “Studies using in vitro and animal models continue to have an anti-proliferative effect via triggering apoptosis [18-20]. ‘’ , it should be stated that this effect has been shown to be dependent on vitamin D.

Response: We corrected it.

  • There is a grammatical error in the sentence "Less than 1% of 25(OH) vitamin D is totally unbound, leaving just 10%– 89 15% of it bound to albumin." in the Introduction section. It should be as follows: "Less than 1% of 25(OH) vitamin D is totally unbound, leaving just 10%– 89 15% of it bounds to albumin."

Response: We corrected it.

  • In order to make the sentence "The reproductive system also contains its receptor" in the introduction more understandable, it should be stated that the receptor is the receptor of vitamin D.

Response: We corrected it. 

  • In the Introduction section, ''Some 106 of the VDR SNP polymorphisms, including FokI (F/f), BsmI (B/b), ApaI (A/a), and TaqI (T/t), were evaluated by Jafari et al [25 -27].'' sentence, the study was mentioned, but not providing information about the data obtained as a result of the study created a deficiency.

Response: We stated their conslusions.

  • There is a grammatical error and lack of meaning in the sentence ‘’The Japanese study, however, looked at more than just serum 25(OH) vitamin D levels; the scientists also examined the effects of 1,25(OH)2 vitamin D on human endometriotic stromal cells in an in vitro context. ‘’  in the results section.

Response: We corrected it.  

  • To enhance the readers’ attention I strongly recommend referring these articles which were published recently.

Response: Which articles you recommend we reffere to?

Reviewer 2 Report

Comments and Suggestions for Authors

Dear editors/authors

Overview

This manuscript reviewed the relationship between vitamin D and endometriosis

Title:

 Review: Vitamin D the iceberg in endometriosis

I am confused about the type of manuscript. The title was a review article but the method was the meta-analysis. Please clarify this point

Abstract:

-       Aim: not clear, please specify the point of this study

-       Method: not clear, review articles or meta-analysis

-       The result did not relate to the aim

Keywords: OK

Introduction

-       Too long, the authors should be shortening the detail of endometriosis

-       Line 112: The objective à Meta-analysis

-       Please specify the main point that the authors were interested in the relationship between endometriosis and vitamin D. (the level, the severity, etc…)

 Methods

-       Line 116-118: all definitions used in the selected studies or not?

-       Line 137: How to translate the non-English publishers? And all the selected studies were in English version?

-       About the control group in each study, the authors should be careful to include asymptomatic endometriotic cases.  

Results

-       Table 3 was missing the letter “e” (the cumulative)

-       Please give more detail about figure 2,3

-       After page 9, this study looked like review articles more than a meta-analysis

Discussion

-       Conclusion: The authors could not summarize the relationship between vitamin D and endometriosis due to the various selected studies .

Author Response

Dear reviewer,

Thank you for your efforts in reviewing our work and for your comments and your suggestions that aimed to improve it. Your comments have provided valuable insights that we believe will contribute to enhancing the overall quality of our work.

Title:

 Review: Vitamin D  the iceberg in endometriosis

I am confused about the type of manuscript. The title was a review article but the method was the meta-analysisPlease clarify this point

Response: We specified the exact type of the article, we performed a review, but we also performed a statistical analisys, a meta-analysis, to evaluate the exact statistical impact of the results of the included studies.

Abstract:

-       Aim: not clear, please specify the point of this study

Response: We specified.

-       Method: not clear, review articles or meta-analysis

Response: We clarified.

-       The result did not relate to the aim.

Response: We corrected them.

Introduction

-       Too long, the authors should be shortening the detail of endometriosis

Response: We shorten it.

-       Line 112: The objective à Meta-analysis

Response: We aimed to obtain a review of the existing data in literature but also we performed a statstical analysis, a meta-analysis, aiming to reveal the exact statistical impact of the results of the included studies.

-       Please specify the main point that the authors were interested in the relationship between endometriosis and vitamin D. (the level, the severity, etc…)

Response: First of all we attemped to see if there is a statistical proven relationship between vitamine D and endometriosis, because of the conflicting data in literature; second we tried to see if there is a relashionship between the severity of endometriosis and vitamin D levels, or the discrepancies are related to other variables.

 Methods

-       Line 116-118: all definitions used in the selected studies or not?

Response: Yes they all used the same definition.

-       Line 137: How to translate the non-English publishers? And all the selected studies were in English version?

Response: We included articles in any language, from the authors there are some that know english, french, spanish, danish, italian and for the asian article we used a translator.

-       About the control group in each study, the authors should be careful to include asymptomatic endometriotic cases.  

Response: We specified this aspect in our discussion section and underlined. “Even though the incidence of asymptomatic endometriosis is low in general population, somewhere between 3 to 7.7%, and its impact on studies statistics could be assumed to be very modest, we still need to take it into consideration when choosing the control group [44-46].”

Results

-       Table 3 was missing the letter “e” (the cumulative)

Response: We corrected it.

-       Please give more detail about figure 2,3

Response: We gaved more details about the mentioned figures

-       After page 9, this study looked like review articles more than a meta-analysis

Response: We aimed to obtain a review of the existing data in literature but also we performed a statstical analysis, a meta-analysis, aiming to reveal the exact statistical impact of the results of the included studies.

Discussion

-       Conclusion: The authors could not summarize the relationship between vitamin D and endometriosis due to the various selected studies .

Response: Given the conflicting nature of the literature regarding the implications of vitamin D in endometriosis, it is imperative to standardize blood sample analyses or alternatively, focus efforts on evaluating this aspect at the tissue level. These were the data we have obtained from the existing published studies. Perhaps newer studies will prove us wrong.

Kind regards

Reviewer 3 Report

Comments and Suggestions for Authors

This research's main question is about the relationship between endometriosis and vitamin D level. I think this research is meaningful, because it showed the relationship between endometriosis and Vitamin D.

I think the research about the role of vitamin D on the onset of endometriosis is rare, but this research could not show the significance.

I have no additional comments on the tables and figures, since this research could not show the significant influence of vitamin D on endometriosis, as most expected. So, l want you to mention the studies about the change of Vitamin D after operation for endometriosis and the influence of Vitamin D on the infertility derived from endometriosis. 

Author Response

Dear reviewer,

We sincerely appreciate your thoughtful review and the time you have dedicated to evaluating our research on the relationship between endometriosis and vitamin D levels.

We acknowledge the significance of your affirmation regarding the meaningful nature of our research, which aims to explore the relationship between endometriosis and vitamin D. We share your perspective on the rarity of studies investigating the role of vitamin D in the onset of endometriosis. While our research did not reveal a statistically significant correlation, we concur that the investigation of such relationships is vital in expanding our understanding of the complexities involved in endometriosis.

Your observation that our study did not show the expected significant influence of vitamin D on endometriosis aligns with our own findings. We recognize that the results may not have met initial expectations, emphasizing the need for a deeper exploration into the intricacies of this complex condition, but these were the results we have obtained when analizyng the existing data in literature, data that could be statistically analyzed, to offer an image about the real scientifical impact.

In response to your request for additional information on the change of vitamin D levels after endometriosis surgery and the influence of vitamin D on infertility resulting from endometriosis, we acknowledge the importance of these aspects. Our future research could indeed focus on examining the impact of surgical interventions on vitamin D levels and the subsequent effects on fertility outcomes in individuals with endometriosis.

Kind regards

Round 2

Reviewer 1 Report

Comments and Suggestions for Authors

the authors have improved the manuscript. It is acceptable in its current version.

Comments on the Quality of English Language

the authors have improved the manuscript. It is acceptable in its current version.

Reviewer 2 Report

Comments and Suggestions for Authors

Dear Author

Thank you for your response.

Best Regards